# UNLEARNING THE UNWANTED DATA FROM A PERSONALIZED RECOMMENDATION MODEL

## ABSTRACT

Recommender Systems (RS) learn user behavior by monitoring their activities on the online platform. In a few scenarios, users consume the content but don't want to get their recommendations because a). They consumed the content by mistake, and those interactions have been utilized in personalizing the model; b) The content was consumed by someone else on their behalf; c) Data acquisition was faulty because of machine failure; d) The user has lost interest in the service, etc. Out of any of these reasons, the user wants the data that was used for generating the recommendation to be unlearned by RS. The constraints with this unlearning are 1) The user's other data should be intact, 2) Personalized experience should not be affected, and 3) We can not afford training from scratch. To solve the stated problem, a few unlearning strategies have already been proposed, but unlearning the matrix factorization-based model is not much explored. In this work, we propose a solution of unlearning from the faulty recommendation model ($m_1$) by diluting the impact of unwanted data. To do so, we first correct the unwanted data and prepare an intermediate tiny model $m_2$, referred to as the rescue model. Further, we apply the convolution fusion function (CFF) on the latent features acquired using $m_1, m_2$. The performance of the proposed method is evaluated on multiple public datasets. We observed that the proposed method outperforms SOTA benchmark models on recommendation tasks.(code[1] files)

## 1 INTRODUCTION

The main objective of recommendation systems is to provide personalized suggestions and assist in making choices for users based on their previous interests. The recommender system plays an important role in people's daily lives by alleviating information overload. Furthermore, personalized decision-making on online platforms is heavily dependent on the recommendations received. Collaborative filtering-based recommendation techniques such as matrix factorization (MF) have been extensively applied for building personalized recommendations (He et al. (2020); Okura et al. (2017)).

Sometimes, the user wants to forget the videos seen or books read at a difficult time and does not want anyone to remind him/her of those things again. EU's General Data Protection Regulation (GDPR) ensures data privacy and security. Under the legal regulation as *right to be forgotten*, one can request online platforms to delete their specified data and associated learnings of the recommendation model.

Conventional intelligent systems work on the historically generated data for predicting near-future outcomes. Such systems do not consider users' privacy or shift in data generation. Sometimes, laws and regulations or changes in user preferences or environmental impact force us to unlearn or forget the old choices learned over a certain timeline. Further, only deleting the data that needs to be forgotten will not solve the problem because the intelligent model trained with the specified data retains the traces of that data and makes predictions based on those traces in the model.

In the context of recommendation systems, the online platforms understand the user's behavior with the help of their activity during engagement with the platform. User's activities on the platform drive the state of the recommendation model, which continually monitors users' behavior and tries to

---

[1] https://anonymous.4open.science/r/UnlearningRecommendation-ICLR

adapt accordingly to provide more relevant and fresh recommendations. The adaptation of the model happens by leveraging the user's behavioral features. The problem arises when the user requests the platform to forget the data acquired by the platform within the specified period, whereas the model has already consumed that data and transited to its new state. The user's request for forgetting the data makes the new state of the model illegitimate. Hence, recommendations given by the model are irrelevant and would lead to a bad user experience. The most common option to get rid of this faulty state is to train the model from scratch by using the entire training data, only eliminating the data requested to unlearn, which is very costly. The training from scratch is not a viable solution because a). It is a time-consuming job b) Previous historical data may not be available. Therefore, we need to obtain a new legitimate state of the model by utilizing only the current illegitimate model state and data to unlearn, which is the motivation of this work.

Recently, multiple studies have explored machine unlearning, but very little work has been done in the area of recommendation unlearning (Li et al. (2022)). The most simple methodology of unlearning is to delete the data that needs to be forgotten and train a new recommendation model using the remaining dataset. However, such an approach is not possible because of the high computational cost in the case of huge datasets. A little work has been done to efficiently unlearn the data from the machine learning models (Bourtoule et al. (2021), Cao & Yang (2015)).

This paper proposes an efficient recommendation unlearning approach that unlearns the requested data and provides intact recommendation performance. To the best of our knowledge, this is the first work that unlearns from the pretrained model using only the specified data to unlearn and without the need of training from scratch. The main contributions of the work can be pointed out as:

- This work proposes a novel method for unlearning the model with the specified data without training from scratch.
- A novel convolution fusion-based framework has been proposed that fuses two different latent features and provides latent features that help in obtaining a full matrix that is equivalent to the first model features that were trained with all historical data.

The paper has been organized as follows. In-depth literature related to recommendation systems and the emergence of recommendation unlearning has been reviewed in the next section 2. Section 4 presents the proposed methodology and notations used. Section 5 includes the results on three different real-world datasets. Finally, section 7 concludes our findings and research with future research directions in the area of recommendation unlearning.

## 2 LITERATURE SURVEY

In literature, unlearning has been analyzed on the parameters of accuracy and time complexity. Generally, machine learning researchers have compared the performance with retraining the model. Our work is related to existing work in machine unlearning and recommendation unlearning. Therefore, we divided the literature survey into two categories.

### 2.1 MACHINE UNLEARNING

Unlearning is an important and new research area that has gained popularity because of privacy laws in the data science domain. The methodologies of deleting the learned knowledge of a machine learning model have been studied in different aspects, such as privacy unlearning (Chen et al. (2021)), and concept drift unlearning (You & Lin (2016)). Most of the studies proposed machine unlearning algorithms in image classification where the objective is to unlearn a particular class of the image dataset (Mehta et al. (2022)). Based on the work done in this area, the concept of unlearning can be broadly categorized as exact unlearning, differential privacy, approximate unlearning, dataset cleaning, mitigation, and continual learning. Bourtoule et al. (2021) and Schelter et al. (2021), presented exact unlearning based methods. In their paper, Bourtoule et al. (2021) had given an ensemble of models on shards of the dataset and Schelter et al. (2021) presented a technique of unlearning for randomized trees. Exact unlearning-based work faces a limitation in size of data deletion. It assumes that such techniques can perform better with deletion requests of very small size of data. Brophy & Lowd (2021) has given a random-forests-based unlearning approach that retrains the subtrees as needed. Ginart et al. (2019) proposed a data elimination technique for k-means clustering.

## 2.2 RECOMMENDATION UNLEARNING

Recommendation unlearning is a new area of research, and we have seen very little work done on it. Chen et al. (2022) introduces an erasable recommendation framework that employs three data partition strategies for balanced shard splitting and an aggregation method to enhance recommendation. Zhang et al. (2023) introduces the Influence Function-based Recommendation Unlearning (IFRU) framework that uses the influence function to quantify the spillover influence of erasing data on the computational graph of recommender models.

## 3 PRELIMINARIES

### 3.1 MATRIX FACTORIZATION

Matrix factorization (MF) (Koren et al. (2009)) is a mathematical technique to work on matrices. It is a powerful method that is widely used in collaborative filtering-based recommendation systems to find hidden user-item interactions by approximations.

Let us consider a matrix $\mathcal{R}$ containing user $u$, items $v$, and the rating $r$. MF approximates $\mathcal{R}$ based on the number of latent factors as the part of two matrices $\mathcal{P}$ (user latent features) and $\mathcal{Q}$ (item latent features). The values of $\mathcal{P}$ and $\mathcal{Q}$ need to be learned through an optimization algorithm to minimize the approximation error. MF works on a user-item rating matrix to uncover the latent features of users and items that enable accurate, personalized recommendations.

## 4 METHODOLOGY

In this section, we first introduce the key notations and formulate the problem for the unlearning task. The proposed methodology for unlearning tasks has been explained in detail.

### 4.1 BASIC NOTATIONS

This section covers the acronyms and notations used in the paper. The users and items are represented as $u$, and $v$, respectively. $r$ is used to show the rating given by the user to a particular item. $\mathcal{R}$ tells about user-item rating matrix. The number of latent features and learning rate are the hyper-parameters that have been represented as $\mathcal{K}$ and $\alpha$. $\mathcal{P}$ represents the user latent features and $\mathcal{Q}$ represents the item latent features.

**Definition 1** *A rating noise is the high values of ratings intentionally set to bring the item in the recommendation list.*

### 4.2 PROBLEM FORMULATION

Let $\mathcal{D}$ be a dataset containing user $u$ and item $i$ interactions as ratings $r$. $\mathcal{D} = \{(u_i, i_j, r_{ij}) \mid u_i$ is a user, $i_j$ is an item, and $r_{ij}$ is the rating given by $u_i$ to $i_j\}$. Here, $u_i$ represents the $i$-th user, $i_j$ represents the $j$-th item, and $r_{ij}$ represents the rating given by the $i$-th user to the $j$-th item. Formally, unlearning problem is identifying a model $\mathcal{RM}'(D_\alpha, \theta)$ such that

$$\mathcal{RM}'(D_\alpha, \theta) = \mathcal{CT}(\mathcal{RM}, D_\gamma) \tag{1}$$

Let $RM(\mathrm{D}, \theta)$, denotes the model with parameters $\theta$ trained with data $d_{train} : d_{train} \subset D$. $\mathcal{CT}$ is a function that performs an unlearning process and produces a model $\mathcal{RM}'$ that has nearly similar performance as the original model $\mathcal{RM}$.

**Theorem 1** *The convolution Fusion Function generates a fading effect.*

**_Proof_** *Let the initial interaction matrix be $\mathcal{D}$, and the faulty interaction matrix be $\mathcal{D}'$ with $users$ in rows and $items$ in columns. Where $\mathcal{D}' = f(\mathcal{D}, \eta, n\_list)$ where $\eta$ is an additive rating noise that randomly assigns a high rating (5 on the scale of 1 to 5) to all the indices of user-item matrix that needs to be unlearned $n\_list$. let*

$$
\begin{aligned}
m_{orig} &= MF(\mathcal{D}) \\
m_1 &= MF(\mathcal{D}')
\end{aligned}
\tag{2}
$$

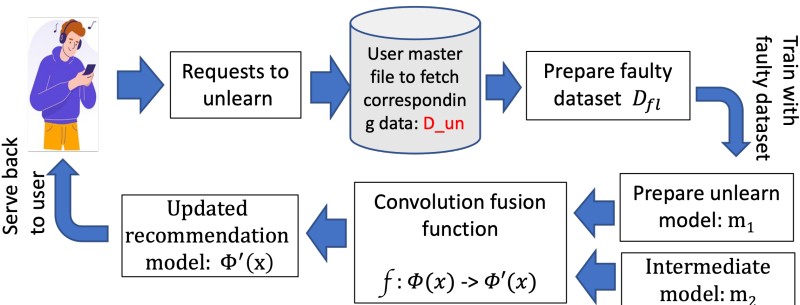

Figure 1: Unlearning Steps
.

$m_1$ is the model that is currently serving the users and needs to unlearn the added noise. If $D_{n\_list}$ is the collection of data that users want to unlearn. A few of columns in $D_{n\_list}$ will be required to unlearn and we set all such entries as ($\mu = \frac{1}{\#users}(\sum_{user=1}^{\#users} item_i)$) which is the average rating of the item and hence we get modified data as $D_{fl_1}$ which only keeps average ratings of the items corresponding to each location of faulty entry.

$$m_2 = MF(\mathcal{D}_{fl_1}) \tag{3}$$

If $U_{lf1}, U_{lf2}, U_{lf3}$ are the user latent features such that $U_{lf1} = m_{orig}.user, U_{lf2} = m1.user, U_{lf3} = m2.user$. If $I_{lf1}, I_{lf2}, I_{lf3}$ are the user latent features such that $I_{lf1} = m_{orig}.user, I_{lf2} = m1.user, I_{lf3} = m2.user$.

$$U_{lf4} = CFF(U_{lf2}, U_{lf3})$$
$$I_{lf4} = CFF(I_{lf2}, I_{lf3}) \tag{4}$$

The CFF function under the Convolution operation $conv(U_{lf2}, U_{lf3})$ considers applying a column filter to both the input matrices. If the value at $i, j$ location at $U_{lf2}, U_{lf3}$ is $\theta_{i,j}, \phi_{i,j}$ respectively, then output of conv will be $(w_1 * \theta_{i,j} + w_2 * \phi_{i,j})$

$$\delta_{U1} = |U_{lf4} - U_{lf1}|, \delta_{I1} = |I_{lf4}| - I_{lf1}|$$
$$\delta_{U2} = |U_{lf4} - U_{lf2}|, \delta_{I2} = |I_{lf4} - I_{lf2}| \tag{5}$$

with the empirical analysis (shown in Table 2b) we observed that $\delta_{U1} < \delta_{U2}$, and $\delta_{I1} < \delta_{I2}$ which simply tells that recovered user latent features $U_{lf4}, I_{lf4}$ is close to the legitimate latent feature $U_{lf1}, I_{lf1}$.

### 4.3 PROPOSED SOLUTION

The detailed steps of the proposed methodology are further explained in this subsection. The paper proposes a novel methodology to unlearn data from a recommendation system. Figure 1 shows the steps to unlearn data from a recommendation system. The proposed solutions are given in the following steps:

#### 4.3.1 SIMULATING THE FAULTY DATA

In our experiments, in the *user-item* interaction matrix, we have randomly selected a few users and their corresponding item ratings (first two columns in Table 3). Since we wanted to simulate the real-world scenario, we added noise to those randomly selected data. Noise is expected to bring those items into the recommendations; hence, in this work, we have set those entries with the highest ratings, which is 5. This noisy data ($D_{fl}$) leads to the model's noisy or illegitimate state ($m_1$), whose recommendations are denied and reported by the user. Therefore, the model is required to be rescued from this state. In a real scenario, the user mentions either the time duration of their activity whose data he doesn't want the platform to consider for personalizing the model, or the user

directly prompts what he doesn't want to see in the recommendations. (It is worth noticing that recommendations are given for those items whose corresponding predicted user ratings are high (eg. 4 or 5)). If $O1$ is the operation of adding noise at random entries in the dataset, as shown in Eq.6.

$$m_{orig} = MF(\mathcal{D}), \quad D_{fl} = O1(D, noise = 5)$$
$$\mathcal{D}' = (D||D_{fl}), \quad m_1 = MF(\mathcal{D}') \tag{6}$$

### 4.3.2 PREPARING THE RESCUE MODEL

The faulty model serving in the previous step leads to faulty recommendations. In this step, we consider the faulty data: $D_{fl}$ (data that leads to a faulty model). This activity of the user has been simulated in the previous step. In this step, we consider the faulty data, prepare a separate model, and consider it as a rescue model ($m_2$). The average rating of those items replaces the faulty entries of the data points. This way, the dataset has a more generic behavior. If $\mu$ is the average rating of each item and $O2$ is the operation to replace the entries of the faulty data with the average rating of the corresponding item (Eq.7).

$$D_{fl_1} = O2(D_{fl}, \mu), \quad m_2 = MF(D_{fl_1}) \tag{7}$$

### 4.3.3 CONVOLUTION FUSION FUNCTION (CFF)

The rescue model is operated over the faulty model to eliminate its faulty features. In this step, we consider the faulty model and rescue model and operate them with CFF. CFF considers the latent features of the faulty model and latent features of the rescue model, and this way, it fades the faulty features with the help of features of the rescue model, as shown in Theorem-1. The behavior is quite intuitive since the rescue model has generic features corresponding to the faulty entries, and fusing both features fades the faulty effect.

$$user\_latent = CFF(m_1.user\_latent, m_2.user\_latent)$$
$$item\_latent = CFF(m_1.item\_latent, m_2.item\_latent) \tag{8}$$

The steps of the convolution fusion function are outlined in Algorithm 1. In Algorithm 1, lines numbers 3 and 4 define the two convolution layers in the neural network, and lines numbers 5 and 6 include two fully connected linear layers. Furthermore, line number 8-20 defines the fusion process of latent features that includes concatenation, reshaping and normalization operations.

### 4.3.4 RESCUED MODEL

The resultant features of the CFF have the least impact of faults (shown in Eq. 8). This process of CFF is performed for user latent features and for item latent features separately; hence, the final matrix is created with the rescued user latent features and rescued item latent features. The rescued model is also known as a transformed model, which is the legitimate state of the model Eq. 9

$$m_f = user\_latent \cdot item\_latent \tag{9}$$

In this work, Matrix Factorization is considered as the recommendation model, and unlearning is realized on the same model. Therefore, the proposed method works on the matrix of user-item and ratings $r_{ij}$ given by a user $u_i$ to item $i_j$. When a user realizes a need to unlearn, he specifies the unwanted data by clicking on certain events or by selecting a timeline. RS unlearn methodology is only required if the requested unwanted data was part of the training data. Based on the user's request, the first check is whether the requested data was part of RS training data previously. If the answer is yes, the current state of the model would be considered as faulty, and data ($D_{fl}$) received in this duration will be used to obtain the legitimate state of the model by unlearning. Faulty model $m_1$ trained using $D_{fl}$ and an intermediate tiny model $m_2$ trained using corrected $D_{un}$. The corrected $D_{un}$ has been formed by replacing the settled highest rating with the average rating value of that particular item. The user latent features $\mathcal{P}_1$, $\mathcal{P}_2$ and item latent features $\mathcal{Q}_1$, $\mathcal{Q}_2$ have been extracted from the model $m_1$ and $m_2$ respectively. Convolution fusion function ($CFF$) accepts the output of $m_1$ and $m_2$ as the input and creates an updated recommendation model that is free from all the patterns and traces of unwanted data. Algorithm 1 shows the process of the fusion function, and components of the fusion function have been demonstrated in figure 2.

In brief, the workflow of unlearning using label aggregation can be given in the following steps:

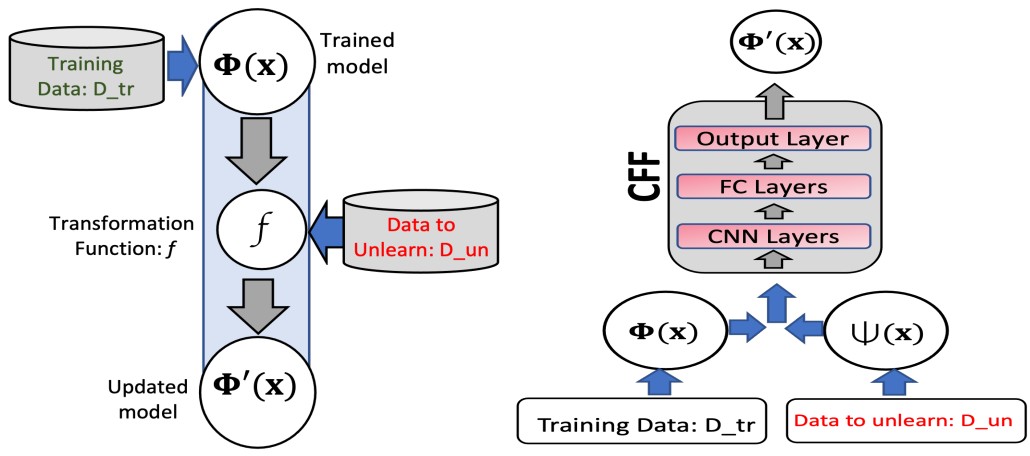

(a) Generic Block of feature unlearning        (b) Fusion function for rating aggregation

Figure 2: The process of feature unlearning and fusion function for rating aggregation.

1. Creating a faulty noisy model $m_1$ using the created noisy dataset $D_{fl}$.

2. Creating an intermediate tiny model $m_2$ using the corrected form of unwanted dataset $D_{fl}$. $D_{fl_1}$ is the fraction of the dataset to be unlearned that is created by replacing the imputed noise with the average rating value of that item.

3. Acquiring the latent features of $m_1$ and $m_2$ to create final unlearned model $M_f$ using fusion function.

---

**Algorithm 1** Convolution Fusion Function

**Input**: $m_1$, $m_2$; $kernel$: kernel size, $type$: type of transfer
**Output**: output tensor $x$

 1: **Function_CF**$(m_1, m_2, type)$:
 2:    $hidden\_dim \leftarrow m_1.latent$
 3:    $conv1 \leftarrow \text{Conv2d}(1, out\_channel, (kernel, 1), \text{stride} = 1)$
 4:    $conv2 \leftarrow \text{Conv2d}(out\_channel, out\_channel2, (1, 1), \text{stride} = 1)$
 5:    $fc1 \leftarrow \text{Linear}(m_1.latent \times 512)$
 6:    $fc2 \leftarrow \text{Linear}(512, m_2.latent)$
 7: **End_Function_CF**
 8: $x_{m_1} \leftarrow Function\_CF(m_1, m_2, user)$
 9: $x_{m_2} \leftarrow Function\_CF(m_1, m_2, item)$
10: $x \leftarrow \text{concatenate}(x_{m_1}, x_{m_2}, \text{dim} = -1)$
11: $x \leftarrow x.\text{view}(-1, 1, 2, x_{m_1}.\text{shape}[-1])$
12: **if** $type$ is "user" **then**
13:    $x = x_{m_1}$
14: **else if** $type$ is "item" **then**
15:    $x = x_{m_2}$
16: **else**
17:     **raise** Error("fusion error")
18: **end if**
19:    $x\_norm \leftarrow (x^2).\text{sum}(\text{dim} = -1).\text{sqrt}()$
20:    $x \leftarrow x/x\_norm.\text{detach}().\text{unsqueeze}(-1)$
21: **return** $x$

---

## 5 EXPERIMENTS

This section discusses the experiments conducted to investigate the performance of our proposed unlearning methodology.

### 5.1 DATASETS

We evaluated our approach using two widely known datasets in recommendation research: Movie-Lens 100K dataset and MovieLens 1M[2] datasets. In Table 1, we report dataset statistics.

Table 1: Dataset Statistics

| Dataset | Users | Items (Movies) | Rating Distribution |
|---|---|---|---|
| Movielens-100K | 943 | 1664 | 1 - 5 |
| Movielens-1M | 6040 | 3706 | 1 - 5 |

### 5.2 EVALUATION METHODOLOGY

In this study, 80% of the user-item utility matrix has been used for the training set, and the rest of the 20% values have been used to test the proposed methodology. Further, Root Mean Square Error has been calculated as a performance metric. In order to check the unlearning efficiency, randomly x% of users have been selected who want to unlearn their y% data. 64 and 0.03 are the selected values of hyper-parameters $\mathcal{K}$ and $\alpha$.

### 5.3 BASELINE

The results of the proposed unlearning methodology have been compared with the following benchmark methods:

***Model Retrain***: This is the naive method to remove unwanted data from a machine learning model. In retraining-based unlearning, all desired samples are removed from the dataset, and the model is re-trained from scratch with the remaining data. Retraining has been considered as the basic benchmark method to remove all unwanted patterns from machine learning models.

***SISA*** Bourtoule et al. (2021): SISA based unlearning is considered to be the state-of-the-art machine unlearning method. In SISA training method, the training data is divided into shards, and further, one model per shard is trained. Finally, the results of all submodels are aggregated via the mean for the final results. When unlearning is requested, a model is trained with the specified shard after removing the unwanted data.

### 5.4 ABLATION STUDY: EFFECT OF UNLEARNED DATA SIZE

This paper proposes a novel unlearning model using the label aggregation method. We perform ablation studies to understand how much data can be forgotten without degrading model performance. This study enables the service provider to set a limit on the size of the data to be forgotten without

---

[2]https://grouplens.org/datasets/movielens/

Table 2

(a) Comparison proposed methodology and existing SISA method on ML-1M

| Parameter | Proposed Methodology | SISA | Retrain |
|---|---|---|---|
| 10, 50 | 1.0235 | 1.14830 | 1.1039 |

(b) Distance between latent features of faulty model and final model with original model

| Matrix Comparison | | $\ell_1$ | $\ell_2$ |
|---|---|---|---|
| 10%-30% | $\mathcal{P}$ | 74.6792 | 73.4687 |
| | $\mathcal{Q}$ | 77.3881 | 76.71722 |
| 20%-50% | $\mathcal{P}$ | 75.3566 | 73.5278 |
| | $\mathcal{Q}$ | 77.2679 | 77.1649 |

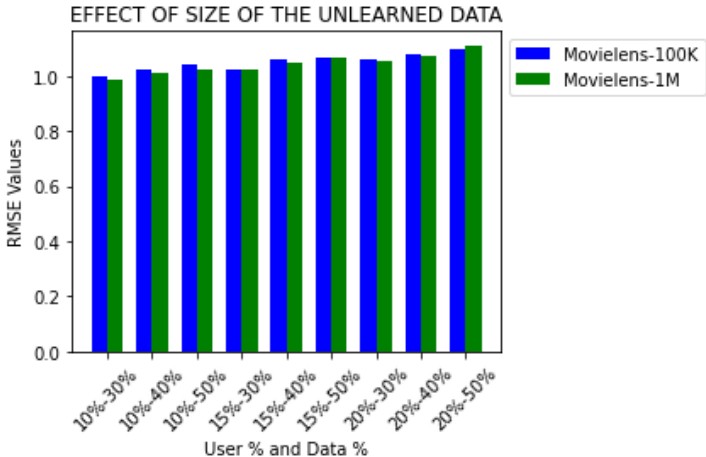

Figure 3: Performance analysis of the proposed methodology with the unlearned data size.

Table 3: Unlearning Data Results for Two Publicly Available Datasets. Results showing the RMSE of the Recommendation Models.

| Users (in %) | F_Data (in %) | ML-100K | | ML-1M | |
|---|---|---|---|---|---|
| | | M1 | Mf | M1 | Mf |
| 10 | 30 | 1.1401 | 0.9960 | 1.1088 | 0.9854 |
| | 40 | 1.1540 | 1.0254 | 1.1206 | 1.0128 |
| | 50 | 1.1682 | 1.0440 | 1.1209 | 1.0235 |
| 15 | 30 | 1.1606 | 1.0259 | 1.1233 | 1.0261 |
| | 40 | 1.1828 | 1.0608 | 1.1333 | 1.0497 |
| | 50 | 1.1699 | 1.0674 | 1.1378 | 1.0639 |
| 20 | 30 | 1.1788 | 1.0632 | 1.1412 | 1.0562 |
| | 40 | 1.1700 | 1.0775 | 1.1413 | 1.0747 |
| | 50 | 1.1793 | 1.0976 | 1.1527 | 1.1077 |

causing much damage to the model's performance. The service provider can set a threshold for model performance to limit the size of data to be unlearned. The study also presents how much data can be unlearned by how many users. Figure 3 presents the results in RMSE on the Movielens-100K and Movielens-1M datasets. We can note two observations from the results shown in figure 3. First, it shows that as we increase the size of the data to be unlearned, the performance decreases slightly. Second, the proposed approach can be used to unlearn up to 50% of the user data without compromising much performance. We see that in cases where small-size data needs to be unlearned, the RMSE is less than 1.

## 6  RESULTS AND ANALYSIS

We present a method that unlearns the data requested by users and retains the original model with the remaining personalized recommendation. This is due to one of the merits of the proposed method that it is not doing any kind of retraining from the scratch. The experimental results have been demonstrated in Table 3. Table 3 shows the performance of the proposed method on the metric Root Mean Square Error (RMSE) on two publicly available datasets. The experiments have also been performed to analyze the performance of the proposed unlearning method on varying data sizes. It shows the performance in different cases when a certain number of users (say x%) request to forget some amount of their data (say y%). RMSE2 presents the error value in recommendation prediction when the model is trained using noisy data. RMSE3 represents the error in the final recommendation model obtained after unlearning the unwanted data. The proposed method uses the matrix factorization method to demonstrate the concept. In the original matrix factorization-based

recommendation model $M_{orig}$, the RMSE values for ML-100K dataset and ML-1M dataset are 1.1005 and 1.1044, respectively. This error value is the original performance of the MF method when the recommendation model learned all historical datasets. When compared with the performance of the original model $M_{orig}$, we see a clear increase in the performance of the final unlearned model $M_f$ in the case of smaller data to be unlearned. As we increase the number of users and their requested data to unlearn, we get a slightly higher RMSE. However, the proposed approach still retains lower or almost equal RMSE than the original MF RMSE in case of large data to be unlearned.

The comparison of RMSE for the proposed methodology and existing unlearning model have been shown in Table 2a. We evaluated the performance of the state-of-the-art unlearning method SISA. In experiments, we fixed the number of shards as 5 and conducted the experiments using Movielens-1M dataset. SISA achieved 1.1483 RMSE when 10% users want to forget 50% of their ratings.

Retrain is a naive method that involves training from scratch after deleting requested data to be unlearned. Therefore, it always shows a lower error. We have taken a similar case for input data as in SISA. Retraining based RMSE is 1.1039 in case 10% of users want to forget 50% of their ratings. As expected, retraining achieves higher performance than SISA and the proposed method is at least as good as retraining. In Table 2a, we see that the proposed approach achieves significantly better performance than both state-of-the-art baseline forgetting techniques.

Table 2b presents the Euclidean distance between the latent features of the faulty model and the final model with the original model. In Table 2b, $\ell_1 = |m_1 - m_f|$ and $\ell_2 = |m_{orig} - m_f|$. We have presented $\ell_1$ and $\ell_2$ in two cases: 1. 10% users request 30% of the ratings need to be forgotten. 2. 20% users wants to remove 50% of their ratings. We see that $\ell_1 > \ell_2$ in all cases. We have noted that the latent features of the final model are closer to the latent features of the original model than the latent features of the faulty model.

## 7 CONCLUSION

Machine unlearning is the new domain of AI that works to improve user experience and helps to have a more engaging experience between the system and users. In this work, we have proposed the need for unlearning in the recommendation system and proposed the methodology to perform it. Our solution neither demands for the previously used training data nor performs training from scratch. The proposed method is quite different from existing methods (e.g. Sharding methods), which are based on the idea of identifying and training a model from scratch among the pool of models. The proposed method uses the already-performing faulty model (which has received training from unwanted data) and data to unlearn (specified by the user). Data to unlearn is assumed to be smaller than the total data; hence, we are preparing a model from that unwanted data separately, and this model helps the faulty model in eliminating the features from the unwanted data. This way, the proposed method is quick and appropriate. We have shown the performance of the proposed method theoretically in Theorem-1 and empirically in Table 3. In this work, we have shown the results on two different datasets and on varying percentages of users and sizes of data. Results have also been compared with baseline SISA and retraining. The proposed method has been proven efficient in terms of performance. The observations of the fusion based recommendation unlearning in combination with rating aggregation are inspirational and motivating to future recommendation unlearning models.

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
