# OpenReview forum: "UNLEARNING THE UNWANTED DATA FROM A PERSONALIZED RECOMMENDATION MODEL"
_ICLR.cc/2024/Conference — ICLR 2024 Conference Withdrawn Submission_

### Official Review · Reviewer_n57L · 2023-11-01

**Soundness:** 3 good
**Presentation:** 1 poor
**Contribution:** 3 good
**Rating:** 3
**Confidence:** 3

**Summary:**

The authors propose a novel method to tackle the problem of unlearning in recommender systems. This is a relevant and critical challenge due to changing user preferences and updating privacy laws and regulations. The proposed method works in the collaborative filtering setting and, in particular, the Matrix Factorization models. The authors propose learning an additional model called the rescue model on a corrected version of the data that has to be unlearned. This is followed by the application of a convolution fusion function (CFF) on the latent space of the previous (faulty) model and the rescue model. The CFF dilutes the impact of the unwanted data, resulting in a final model where the latent features are closer to the original model (before the faulty preferences were learned). The authors compare their approach with two existing unlearning approaches and find an improvement in RMSE. They also evaluate their approach on various combinations of user and data percentages for which the unlearning process has to be done.

**Strengths:**

1. The paper addresses a critical and relevant problem which is not explored much in the recommendation systems domain
2. The authors propose a unique and novel approach for the unlearning problem in the context of recommendation systems
3. The proposed approach seems to be more efficient and computationally feasible than the existing mentioned methods
4. The authors have experimented with various combinations of user and data % to analyze the effect of unlearning using their approach. 5. They have also compared their method with existing unlearning approaches.

**Weaknesses:**

Significant improvements need to be made in the paper writing, style and structure. Concepts need to be properly defined and equation variables explained. E.g.-
1. In section 3.1, R is not mathematically defined in terms of u, v and r.
2. In equation 1, what are $\alpha$ and $\gamma$ in D$\alpha$/D$\gamma$? $\alpha$ was also previously defined as the learning rate hyper-parameter.
3. In theorem 1, motivate the need for $\eta$ (to approximate or simulate the faulty preference to items) or give more context.
4. Why is the definition of $U_{lf1}$, $U_{lf2}$ and $U_{lf3}$ the same as the definition of $I_{lf1}$, $I_{lf2}$ and $I_{lf3}$?
5. Two different definitions of D’ are provided in sections 4.2 and 4.3.1
6. What does D’ = (D||Dfl) mean in equation 6?
7. While there is an improvement in RMSE for the proposed method compared to the baselines, the magnitude of improvement is not very convincing. Can additional metrics like AUC, NDCG etc. be used for evaluation?

**Questions:**

1. Is $n_{list}$ for a particular user or all users? What are the dimensions of $D{n_list}$?
2. What is $D_{fl}$ in figure 1? (referred to as the faulty dataset) Is it referring to $D{n_list}$ or $D_{fl1}$?
3. Why is the definition of $U_{lf1}$, $U_{lf2}$ and $U_{lf3}$ the same as the definition of $I_{lf1}$, $I_{lf2}$ and $I_{lf3}$?
4. What does D’ = (D||Dfl) mean in equation 6?
5. In section 4.3.4, what is $D_{un}$? Is it the same as $D_{fl1}$? If so, why is a different term used?
6. In section 5.2, how were the values 64 and 0.03 selected for the hyper parameters? Is it expected that these values will be the same for both datasets?
7. In section 6, where are RMSE2 and RMSE3 in table 3? What are M1 and Mf in table 3?

---

> ### Author Response · Authors · 2023-11-23
> **Response to reviewer n57L**
>
> We extend our sincere appreciation to the reviewers for their diligent commitment of time and effort in conducting a thorough review.
>
> Ans1. $n_{list}$ is the listed interaction that users want to unlearn, it is the collection of interaction from all the users which is required to unlearn. its a list of indices (row, column) for $user_1$ it is ${(row_1, col_2, )(row_1, col_8),...(row_1, col_k)}$, for $user_2$ it is ${(row_2, col_3),(row_2, col_9),...(row_2, col_k)}$ ...etc.
>
> Ans2. $D_{fl}$ is Figure1 is same as $D^1$ in theorem-1; thanks for pointing it out, we have corrected and made them uniform across the manuscript.
>
> Ans3. Thanks for correcting.
> Updated lines Theorem-1 are as follows:
> If $U_{lf1}, U_{lf2}, U_{lf3}$ are the user latent features such that $U_{lf1}=m_{orig}.user, U_{lf2}=m1.user, U_{lf3}=m2.user$.
> If $I_{lf1}, I_{lf2}, I_{lf3}$ are the item latent features such that $I_{lf1}=m_{orig}.item, I_{lf2}=m1.item, I_{lf3}=m2.item$.
>
> Ans4. $D_{fl}$ is the noisy data, $D$ is the original data, and $D'$ is the data after adding noisy entries (of $D_{fl}$) to the original data $D$. we have updated the equation and made it clear by replacing $D'= (D || D_{fl})$ with $D'= f(D, D_{fl})$ where $f$ is function that replaces entries of $D$ by considering $D_{fl}$.
>
> Ans5. Yes, $D_{un}$ is same as $D_{fl1}$, thanks for pointing it out. We have made them uniform in the manuscript.
>
> Ans6. $K$ is the number of latent features and $\alpha$ is the learning rate, we have added detail in the manuscript. Both parameters are found to be the same in both datasets after tuning.
>
> Ans7. $RMSE2= RMSE(M_1)$, $RMSE3= RMSE(M_f)$, thanks for pointing it out, we have updated this in Table 3 and the corresponding text in the manuscript.

---

### Official Review · Reviewer_o6ud · 2023-11-01

**Soundness:** 1 poor
**Presentation:** 1 poor
**Contribution:** 2 fair
**Rating:** 3
**Confidence:** 4

**Summary:**

This paper underscores the significance of the "right to be forgotten" in recommendations and, in line with this, investigates the challenge of recommendation unlearning. The authors delve into the critical elements of recommendation unlearning, particularly focusing on time efficiency and the possible absence of complete data. They endeavor to facilitate unlearning from a pre-trained model by utilizing only the data that needs to be removed and eliminating the need for starting from scratch. In pursuit of this objective, the authors introduce a convolution fusion-based unlearning framework.

**Strengths:**

S1: The utilization of convolution fusion for achieving unlearning appears to be a novel approach.
S2: The proposed method exclusively leverages the unlearning data and eliminates the necessity for a complete retraining from the ground up.

**Weaknesses:**

W1: In the context of unlearning, the most critical objective is the effective removal of unwanted data from a trained model. Unfortunately, the paper doesn't provide a clear guarantee in this regard. The only supporting evidence seems to be Theorem 1, but it's unclear how the theorem ensures the removal of unwanted data. It appears that the theorem can only justify that a model is closer to "the legitimate one" than another model.

W2: The rationale behind how the convolution fusion function can derive the unlearned model based on the faulty and rescue models is not adequately explained.

W3: The paper makes use of numerous symbols, but some remain unexplained. For instance, "D_γ" in equation (1) and "D||D_fl" could benefit from clarification.

W4: Certain claims in the paper may be overstated. The authors claim that their work is the first to unlearn from a pre-trained model using only specified data for unlearning without the need for training from scratch. However, the cited paper "Recommendation unlearning via influence function" also possesses these characteristics.

W5: There is a noticeable omission of related recommendation unlearning works, such as references [1,2,3].

W6: The experiments conducted in the paper are insufficient to validate the effectiveness of the proposed methods. In the context of recommendation unlearning, there are typically three key objectives, as discussed in the authors' cited papers: 1) unlearning efficacy (ensuring the removal of unwanted data), 2) unlearning efficiency, and 3) recommendation performance. The authors appear to have overlooked the first two aspects in their evaluation.

[1] Li, Yuyuan, et al. "Selective and collaborative influence function for efficient recommendation unlearning." Expert Systems with Applications 234 (2023): 121025.
[2] Yuan, Wei, et al. "Federated unlearning for on-device recommendation." Proceedings of the Sixteenth ACM International Conference on Web Search and Data Mining. 2023.
[3] Schelter, Sebastian, Mozhdeh Ariannezhad, and Maarten de Rijke. "Forget Me Now: Fast and Exact Unlearning in Neighborhood-based Recommendation." SIGIR 2023.

**Questions:**

How does the proposed method ensure the effective removal of unwanted data?

---

> ### Author Response · Authors · 2023-11-23
> **Response to reviewer o6ud**
>
> We express our heartfelt gratitude to the reviewers for their dedicated effort and time devoted to conducting a thorough and detailed review.
>
> As per the reviewer's suggestion, we have rewritten the methodology and included more explanation of the logic behind it to show how the convolution fusion function can derive unlearned models based on faulty and rescue models. As per suggestion, we have also improved the use of symbols along with their explanation. The stated related work has also been incorporated in the modified manuscript.
>
> Our Proposed unlearning methodology is built on top of matrix factorization and uses the convolution fusion function to fade the unwanted data. However, cited paper “Recommendation unlearning via influence function” includes graph computation. Graph Computation itself is a specific technique that is not yet suitable to address the recommendations when participating users and or items are in millions; that was the reason of not including this work in comparison.
>
> Q. How does the proposed method ensure the effective removal of unwanted data?
> Ans. The request of unlearning arises when the model produces recommendations which is unwanted for the user, and it is the testing data that validates the outcome of the model. Therefore, if the unlearning process works well, then it should have modified the features in a way that would lead to the removal of unwanted recommendations, thereby improving the testing performance.

---

### Official Review · Reviewer_GJxA · 2023-11-02

**Soundness:** 2 fair
**Presentation:** 2 fair
**Contribution:** 3 good
**Rating:** 5
**Confidence:** 2

**Summary:**

The authors propose an unlearning approach without training from scratch in the recommendation system. By fusing two different latent features in a convolution fusion-based framework, we can obtain an unlearning model approximate to the retrained model. Experiment results show the convolution fusion-based framework outperforms other baseline models on two public datasets.

**Strengths:**

1. The authors propose a novel unlearning approach in recommendation systems. By applying the convolution fusion function, the data in recommendation systems can be unlearned. Furthermore, the authors also provide proof to show that the convolution fusion function actually generates a fading effect, and this effect can be used to unlearn the data. The approach is simple but effective in matrix factorization-based recommendation systems.
2. Experiments show that the proposed approach outperforms SISA and Retrain in two public datasets.

**Weaknesses:**

1. In Section 2.2, the authors mention the work of Chen et al. (2022) and Zhang et al. (2023), which also study the recommendation unlearning. However, there is no comparison with these methods in the experiment section. Are there any reasons why the authors do not compare these two works? Otherwise, the authors can try to compare with those previous studies to make the results more convincing.
2. In Table 2(a), the authors only show that the proposed approach is better than SISA and Retrain if 10% of users want to forget 50% of their ratings. The experiments would be more convincing if the authors could show more different settings in Table 2(a). For example, 10% of users want to forget 30% of their ratings, or 30% want to forget 10% of their ratings.
3. The convolution fusion function can only be applied in matrix factorization. The feasibility of the proposed approach to the neural-network-based recommendation systems, which are more popular now, is not discussed in the paper.

**Questions:**

1. Why are there no comparisons between proposed methods with previous important baselines, such as Chen et al. (2022) and Zhang et al. (2023)?
2. Why is there only one parameter setting in Table 2(a)? Can the authors add more settings and compare the results?

---

> ### Author Response · Authors · 2023-11-23
> **Response to reviewer GJxA**
>
> We sincerely appreciate the dedication and time invested by the reviewers in conducting a thorough and detailed review.
> Q. Why are there no comparisons between proposed methods with previous important baselines, such as Chen et al. (2022) and Zhang et al. (2023)?
> Ans.
> Chen et al.  Proposed RecEraser, which is based on partition and aggregation, this method always involves training from scratch concepts (with relatively smaller data chunks to prepare the candidate model to participate in aggregation). Whereas the proposed method is based on the motivation of not performing the training from scratch at all, this was the reason for not comparing both methods together because we have already included SISA (sharding-based) in our comparison.
>
> Zhang et al. created a computational graph and erased edges there to realize the unlearning, which is seemingly not a suitable approach to compare with gradient descent based Matrix Factorization, that was the reason for not including their result. However, SISA and Retraining has been unanimously considered baseline by all the peer studies of unlearning.
>
>
> Q. Why is there only one parameter setting in Table 2(a)? Can the authors add more settings and compare the results?
> Ans.  As shown in Table 2(a) we checked the performance when 10% of users unlearn 50% of their data. As more users request to unlearn harder, the impact comes on the model performance hence in order to keep the performance legitimate among all the existing models, we selected 50% of the data of just 10% of users. However, the proposed method has been experimented on nine different settings. As per the reviewer's suggestion, we can easily add the results of other methods on multiple settings.

---

### Official Review · Reviewer_HcxE · 2023-11-05

**Soundness:** 1 poor
**Presentation:** 2 fair
**Contribution:** 1 poor
**Rating:** 3
**Confidence:** 3

**Summary:**

In this paper the authors propose an unlearning method for recommendation systems based on matrix factorization. The method achieves unlearning by diluting the impact of unwanted data. Empirical studies can been done on Movielens-100K and Movielens-1M data to show the efficacy of the proposed method.

**Strengths:**

The paper proposes a new method for matrix factorization unlearning which is not studied by prior research works.

**Weaknesses:**

* The presentation of the paper is poor. The problem itself is defined with vague descriptive languages rather than rigorous math languages. Theorem 1 doesn't make sense to first time readers as either "convolution Fusion Function" or the "fading effect" are not defined. I still don't know what the author tries to show even after the "proof" of the theorem. Besides, you cannot prove a theorem by
"empirical analysis".  Overall the paper reads awkwardly and very confusing even for someone familiar with the context. I would suggest the authors to refactor the paper and have the notations and the problem settings properly defined before going into methods.

* The experiments and evaluation metrics are also confusing. The author conducts experiment comparing the proposed method with re-training and SISA. However, the only metrics reported is the RMSE on the test data. While the proposed method has lower RMSE than both baselines, I am not convinced the proposed model is a better approach: How do you guarantee the unwanted data is unlearned? What the evaluation metric showing the new model does not contain unwanted info? Without such guarantee, isn't the model trained on full data always outperforming ones trained on partial data (i.e. retrain and SISA)?

**Questions:**

* What's 'fading effect' in Theorem 1? Can you define it?
* Section 4.3.4 "hence, the final matrix...", what's the final matrix?
* Does the unwanted data span across the training & test split or only in training?
* How do you evaluate if the unlearned model contains any info from the unwanted data?

---

> ### Author Response · Authors · 2023-11-23
> **Response to reviewer HcxE**
>
> We deeply acknowledge the effort and time given by the reviewers for a detailed review of the manuscript.  As per the suggestion, we have made the following changes in the manuscript:
> 1. Presentation style has been improved for better readability.
> 2. Proper clarifications have been added based on the reviewers' comments.
> 3. Sentences are reframed into shorter ones with mathematical language (including problem setting) to improve the quality of the article.
>
> Our responses to the questions are as follows:
>
> Q. What's 'fading effect' in Theorem 1? Can you define it?
> Ans. The fading effect is a smooth real-time change in the input dataset. In the context of the Convolution Fusion Function, the Fading effect is the removal of a specific outcome realized because of faulty data (which is required to unlearn). The definition has been added in the manuscript.
>
> Q. Section 4.3.4 "hence, the final matrix...", what's the final matrix?
> Ans. The final matrix is the improved user-item interaction matrix, which is regained after going through CFF based unlearning process. The final matrix is the outcome of the unlearning process and it does not contain the data that the user requested to unlearn.
>
>
> Q. Does the unwanted data span across the training & test split or only in training?
> Ans. The requirement of unlearning comes only when the model receives training with the data that was erroneous (considering a continual learning paradigm). An erroneous data leads to a faulty state of the model, and we need to rescue the model from this state. At the same time, test data has always been unseen and is only used to validate the state of the model (typical case of any ML model). Hence, we propose that faulty data is always part of training data only.
>
> Q. How do you evaluate if the unlearned model contains any info from the unwanted data?
> Ans. The request of unlearning arises when the model produces recommendations which is unwanted for the user, and it is the testing data that validates the outcome of the model. Therefore, if the unlearning process works well, then it should have modified the features in a way that would lead to the removal of unwanted recommendations, thereby improving the testing performance.

---

### Meta-Review · Area_Chair_8wXA · 2023-12-10

**Metareview:**

I recommend to reject this paper.

  In this paper the authors proposed an unlearning method for recommendation systems based on matrix factorization such that the training from scratch is not needed to unlearn some of the user-user actions. However, as pointed by reviewers, the presentation/writing of this paper is poor and the experiments conducted in the paper are insufficient to validate the effectiveness of the proposed methods for unlearning efficacy and unlearning efficiency. As a result, I recommend to reject this paper.

**Justification For Why Not Higher Score:**

N/A.

**Justification For Why Not Lower Score:**

N/A

---

### Decision · Program_Chairs · 2024-01-16

Reject